# Nutritional Ultrasonography, a Method to Evaluate Muscle Mass and Quality in Morphofunctional Assessment of Disease Related Malnutrition

**DOI:** 10.3390/nu15183923

**Published:** 2023-09-09

**Authors:** Juan José López-Gómez, David García-Beneitez, Rebeca Jiménez-Sahagún, Olatz Izaola-Jauregui, David Primo-Martín, Beatriz Ramos-Bachiller, Emilia Gómez-Hoyos, Esther Delgado-García, Paloma Pérez-López, Daniel A. De Luis-Román

**Affiliations:** 1Servicio de Endocrinología y Nutrición, Hospital Clínico Universitario de Valladolid, 47003 Valladolid, Spain; rebebel9@gmail.com (R.J.-S.); oizaolaj@saludcastillayleon.es (O.I.-J.); dprimoma@saludcastillayleon.es (D.P.-M.); bramosb@saludcastillayleon.es (B.R.-B.); emiliagomezhoyos@gmail.com (E.G.-H.); delgadogarciaesther@gmail.com (E.D.-G.); palomaperezz18@gmail.com (P.P.-L.); dadluis@yahoo.es (D.A.D.L.-R.); 2Centro de Investigación Endocrinología y Nutrición, Universidad de Valladolid, 47003 Valladolid, Spain; 3Facultad de Medicina, Universidad de Medicina, 47003 Valladolid, Spain

**Keywords:** nutritional ultrasonography, disease-related malnutrition, morphofunctional assessment, echogenicity

## Abstract

Nutritional ultrasonography is an emerging technique for measuring muscle mass and quality. The study aimed to evaluate the relationship between the parameters of body mass and quality of ultrasonography with other parameters of morphofunctional assessment in patients with disease-related malnutrition (DRM). Methods: A cross-sectional study was developed on 144 patients diagnosed with DRM according to the Global Leadership Initiative on Malnutrition (GLIM) criteria. Morphofunctional evaluation was assessed with anthropometric variables, handgrip strength and bioelectrical impedanciometry (BIA). Nutritional ultrasonography of quadriceps rectus femoris (QRF) was made (muscle mass (Muscle Area of Rectus Femoris index (MARFI)), Y axis and muscle quality (X-Y index and echogenicity). Results: The mean age of patients was 61.4 (17.34) years. The prevalence of sarcopenia in the sample was 33.3%. Patients with sarcopenia (S) had lower values of MARFI [(S: 1.09 (0.39) cm^2^/m^2^; NoS: 1.27 (0.45); *p* = 0.02), Y axis (S: 0.88 (0.27); NoS: 1.19 (0.60); *p* < 0.01) and X-Y index (S: 1.52 (0.61); NoS: 1.30 (0.53); *p* < 0.01)]. There was a correlation between BIA parameters (phase angle) and muscle mass ultrasonographic variables (MARFI) (r = 0.35; *p* < 0.01); there was an inverse correlation between muscle quality ultrasonographic variables (echogenicity) and handgrip strength (r = −0.36; *p* < 0.01). In the multivariate analysis adjusted by age, the highest quartile of the X-Y index had more risk of death OR: 4.54 CI95% (1.11–18.47). Conclusions: In patients with DRM and sarcopenia, standardized muscle mass and muscle quality parameters determined by ultrasonography of QRF are worse than in patients without sarcopenia. Muscle quality parameters had an inverse correlation with electric parameters from BIA and muscle strength. The highest quartile of the X-Y index determined by ultrasonography was associated with increased mortality risk.

## 1. Introduction

Disease-related malnutrition (DRM) is a highly prevalent pathology which has become a significant challenge in our health system. This disease has a prevalence between 20% and 50% in hospitalized patients [1,2]. The presence of this situation can be associated with an increase in complications and mortality. The EFFORT study showed that patients with malnutrition diagnosed by GLIM criteria had more risk for adverse clinical outcomes (OR: 1.53; 95%CI: 1.22–1.93) [3]. This condition may also increase the cost of hospitalization; in this way, the patients with the risk of malnutrition are supposed to have a high cost during hospitalization [4].

Malnutrition can be associated with other conditions, such as sarcopenia, defined by a loss of muscle mass and function. This disease was described as a primary condition associated with aging and frailty, but in 2019 the European Working Group on Sarcopenia in Older People (EWGSOP2) raised the secondary sarcopenia associated with several diseases [5]. This pathology can be present in up to 15% of patients with malnutrition and 32% of patients with cachexia in older adults [6]. The presence of sarcopenia also increases the risk of complications in surgical [7], medical patients [8,9] and older adults [10].

The main societies in nutrition worldwide, like the European Society of Clinical Nutrition and Metabolism (ESPEN) and the American Society of Parenteral and Enteral Nutrition (ASPEN), recommend starting medical nutrition treatment in medical and surgical patients at risk of malnutrition [2]. Therefore, an adequate and early diagnosis of malnutrition is very important to carry out an adapted Medical Nutrition Therapy to prevent complications [11]. 

The adequate diagnosis of malnutrition and sarcopenia is based on some tests to evaluate dietary intake, body composition, muscle strength and function, and biochemical parameters. This global approach to diagnose malnutrition has been called morphofunctional assessment of disease-related malnutrition [12]. Morphofunctional Assessment can help us evaluate patients at risk of malnutrition and an early diagnosis of disease-related malnutrition for personalized treatment.

The evaluation of body composition, especially muscle mass, is an important component of the diagnosis of malnutrition and sarcopenia, and it plays an essential role in monitoring the nutritional treatment of DRM. Nevertheless, the diagnosis of muscle quantity and quality is difficult. Some techniques are not as accurate as anthropometry with perimeters or estimative equations based on bioimpedanciometry. Besides, there are some tests like computerized tomography (CT), or magnetic resonance imaging (MRI) considered the gold standard but more expensive and not feasible in routine clinical practice [13]. 

Nutritional ultrasonography is an emerging technique in diagnosing DRM and sarcopenia to measure muscle mass and quality [14]. This probe allows a simple method to evaluate muscle mass in the consultation or bedside in hospitalized patients. It is an economical and not invasive test, and it helps us to determine several muscular groups. The main limitation of this technique is the scarce evidence of its relationship to the prognosis of DRM, the lack of use of a standardized muscle group and the need for validation with cutoff points for DRM and sarcopenia. Finally, this technique needs trained personnel capable of performing this ultrasound method and managing the data on the software [15].

Nutritional ultrasonography allows us to measure muscle mass as a quantitative method by determining muscle thickness and muscle area. A study by Fischer et al. in 2022 probed that ultrasound can predict CT L3 skeletal muscle area (SMA) [16]. On the other hand, muscle ultrasonography helps us to evaluate the quality of muscle by measuring its shape and echogenicity. A study in oncologic patients shows that ultrasonography correlates with body composition techniques with functional components such as bioimpedanciometry (phase angle) and handgrip strength [17].

Nutritional ultrasonography offers us an economical, feasible and not harmful technique to assess muscle mass and quality. This method of study of body composition allows us to make an early diagnosis of malnutrition to personalize medical nutrition therapy. Besides, the follow-up of changes in ultrasonography can help to monitor the effect on muscle of nutritional treatment.

This study aimed to evaluate the feasibility of nutritional ultrasonography in diagnosing malnutrition and sarcopenia and its relationship with the prognosis of patients with DRM. The main objectives of the study were to evaluate the relationship between the parameters of body mass and quality of ultrasonography with techniques of body composition such as bioimpedanciometry and muscle quality determined by handgrip strength, to describe the differences in muscle mass determined by ultrasonography in the function of diagnosis of sarcopenia and to characterize the prognosis of basic pathology related to the ultrasonography parameters.

## 2. Materials and Methods

### 2.1. Study Design

A cross-sectional study was developed in 144 patients diagnosed with disease-related malnutrition with GLIM criteria [18]. The patients were recruited in the Clinical Nutrition Unit of Clinic Universitary Hospital of Valladolid between January 2021 and September 2022.

After signing informed consent, patients were interviewed about medical history, disease progression and nutritional anamnesis. It was done anthropometry, electric bioimpedanciometry, handgrip strength and muscle mass and quality were evaluated by nutritional ultrasonography.

The study was approved by the ethics committee of East Valladolid Area with code PI 22-907 and carried out following the principles of the Helsinki Declaration.

### 2.2. Study Subject

The selected patients had the following inclusion criteria: community patients with a diagnosis of disease-related malnutrition with GLIM criteria; over 18 years. The exclusion criteria were: Uncontrolled hepatopathy, chronic kidney disease over the IV stage, and patients who didn’t sign informed consent.

### 2.3. Variables

Anthropometry: The anthropometric variables measured were weight (kg), height (m); body mass index (BMI) as weight/height × height (kg/m^2^); percentage of weight loss (%TWL): (Usual weight (kg) − Actual weight (kg))/Usual weight (kg) × 100). Arm Circumference (AC) (cm) and calf circumference (CC) cm were measured using the guideline of “Anthropometric variables of the Spanish sports population”, which uses a modified version of the International Society for the Advancement Kinanthropometry (ISAK) protocol. The arm circumference was made at the middle point between the acromium and radium head with a relaxed arm. The calf circumference was made with the patient standing at the maximum perimeter between the knee and ankle [19]. One was taken measured at the right member (arm and calf). The person who did anthropometry was a dietitian-nutritionist formed in anthropometric measurement with skills in nutritional assessment and anthropometry. The measurements were always taken by the same operator.

Muscle Function: The muscle function was obtained with handgrip strength (JAMAR^®^ dynamometer, Preston, Jackson, Missouri, MO, USA). The measure was taken with the patient sitting with the dominant arm at a straight angle with the body. We made three determinations, and we chose the highest value.

Body composition:-Bioelectrical Impedanciometry (BIA): The BIA measure the hydration and cell density of the body by the determination of electric parameters such as resistance, reactance, and phase angle. The use of validated estimative equations allows us to define the compartments of body composition [20]. Bioimpedanciometry (BIA NutriLab; EFG Akern, Akern, Pisa, Italy) was performed between 8:00 and 9:15, after an overnight fast and after a time of 15 min in the supine position. The BIA measured the parameters of impedance (Z), resistance (R) and capacitance (X). The phase angle (PhA) is calculated with: PhA = ((X/R) × 180°/π). It was calculated by estimative equation fat mass (FM), fat-free mass (FFM), fat-free mass index (FFMI) and percentage of skeletal muscle mass (%MM) [20]. We estimated the appendicular skeletal muscle mass (ASMI) by Sergi Formula: −3.964 + (0.227 × RI) + (0.095 × weight) + (1.384 × sex) + (0.064 × Z), where RI resistivity index (sex: Male = 1; Female = 0) [21].-Nutritional Ultrasonography: We made a muscular ultrasonography of the quadriceps rectus femoris (QRF) of the dominant lower extremity with a 10 to 12 MHz probe and a multifrequency linear matrix (Mindray Z60, Madrid, Spain). The measurement was made with the patient in the supine position. The probe was aligned perpendicular to the longitudinal and transverse axis of QRF. The determination was performed without compression at the level of the lower third from the superior pole of the patella and the anterior superior iliac spine [14].

The variables that we measured to assess muscle mass were the anteroposterior (Y) and transversal muscle thickness (X), cross-sectional muscle area (MARF) and muscle circumference (MCRF) [15]. The area was standardized by height (muscle area (cm^2^)/height × height (m^2^) and is named the muscle area rectus femoris index (MARFI). The variables used to assess muscle quality were X-Y index ((Xaxis/Yaxis)/height^2^) that relate transversal and anteroposterior muscle thickness; on the other hand, we measured muscle echogenicity with Image J software, version 1.52p (National Institutes of Health (NIH), Bethesda, MD, USA) [22]; to display echogenicity, we consider 0 as complete black color and 255 as complete white color, we selected a region of interest (ROI) centered in QRF, and we take the median of the values. We standardize by the formula: (Median/255) × 100 (see Figure 1).

Diagnostic test: Severity GLIM criteria: It was used to determine the severity GLIM criteria of severity to characterize the type of malnutrition (mild or severe). We considered severe malnutrition for those with phenotypic criteria of more than 10% weight loss in the last six months or >20% in one year or a BMI < 18.5 kg/m^2^ in <70 years or <20 kg/m^2^ in >70 years [18].

EWGSOP2 criteria: To determine the diagnosis of sarcopenia, we used the EWGSOP2 criteria [5]. Low muscle strength (or dynapenia) was considered as a handgrip strength <16 kg in women and <27 kg in men; low muscle mass was considered with appendicular skeletal mass index (ASMI) determined by BIA (ASMI < 5.5 kg/m^2^ in women and ASMI < 7 kg/m^2^ in men).

Comorbidity and mortality:

We consider the morbidity of disease, the number of visits to emergency service, the number of hospitalization episodes and the death.

### 2.4. Statistical Analysis

The database has been registered with permission of the National Data Protection Agency. The collected data was stored in a database using the statistical software SPSS 23.0 (SPSS Inc., Chicago, IL, USA).

Continuous variables were presented as mean and standard deviation, while parametric variables were analyzed using the unpaired Student’s *t*-test. For non-parametric variables, tests such as Friedman, Wilcoxon and Mann-Whitney U test will be used. To compare variables in more than two groups, the ANOVA U test was applied with the Bonferroni post-hoc test. The analysis of the variables at different times of the study was carried out using multivariate analysis of variance (MANOVA). Qualitative variables were expressed as percentages and analyzed using the Chi-square test, with Fisher and Yates adjustments when necessary. Statistical significance was considered as a *p*-value with a value below 0.05.

## 3. Results

### 3.1. Sample Description

It analyzed 144 patients diagnosed with disease-related malnutrition (DRE). 60.4% of patients were women, and the average age was 61.4 (17.34). The pathologies which cause malnutrition are represented in Figure 2.

The prevalence of sarcopenia with EWGSOP2 criteria was 33.30%. The low muscle mass criterion was fulfilled in 45.8% of patients, and the low muscle strength (dynapenia) was fulfilled in 51.4%. There were no differences in sarcopenia (*p* = 0.72) or dynapenia (*p* = 0.12) between sexes, but there were differences in low muscle mass criteria (*p* < 0.01) (Figure 3).

Morphofunctional assessment variables and the differences between sexes are represented in Table 1.

The morbidity registered at three months was nine deaths (6.3%), 40 patients (27.8%) were hospitalized at least one time, and 70 patients (48.7%) went to emergency services at least one time. Between admitted patients, the median of admissions was 1 (1–2) times, and the days of admission were 10 (5–18.75) days. Between those who were admitted, the median of visits to emergency services was 1 (1–2.25) times.

### 3.2. Morphofunctional Assessment and Diagnosis of Sarcopenia

We compared the variables of morphofunctional assessment in the function of diagnosis of sarcopenia. We observed significant differences in anthropometry and bioimpedanciometry parameters except for reactance (Table 2). If we compare the ultrasonography parameters, we observe differences in muscle area as a measure of muscle mass and X-Y index as a quality measure (Table 2).

After the stratification in the function of components of sarcopenia (dynapenia and low muscle mass), we have observed differences in lower values of MARFI in those patients with dynapenia and higher values of echogenicity in these patients. We didn’t observe these differences in patients with low muscle mass; the only difference observed is a lower value of echogenicity in patients with low muscle mass (Table 3). If we compare BIA parameters, the differences were in functional parameters such as reactance and phase angle in those patients with dynapenia, and there were differences in all parameters of BIA in those with low muscle mass (Table 3).

### 3.3. Comparison of Parameters of Muscle Mass and Quality of Nutritional Ultrasonography

We considered muscle mass parameters in nutritional ultrasonography, the muscle area of the rectus femoris index (MARFI) and the muscle circumference of the rectus femoris index (MCRFI). We also considered muscle quality parameters, echogenicity, and X-Y index. It was observed a positive correlation between quality parameters (X-Y index and echogenicity) (r = 0.27; *p* = 0.03) and between muscle mass parameters (MARFI and MCRFI) (r = 0.75; *p* < 0.01) (Figure 4). When we compared quality and muscle mass parameters, we observed a positive correlation between MCRFI and the X-Y index (r = 0.22; *p* = 0.01) and a negative correlation between the MARFI and X-Y index (r = −0.30; *p* < 0.01).

### 3.4. Comparison of Parameters of Muscle Mass and Quality of Nutritional Ultrasonog-Raphy

We compare variables obtained from nutritional ultrasonography with variables of body composition (BIA), muscle strength (handgrip strength) and anthropometry (braquial and calf circumferences). A positive correlation was observed between muscle mass parameter MARFI and body composition parameters such as ASMI, MMI and Phase Angle; a negative correlation was observed between MARFI and electric parameters from BIA (resistance and phase angle) (Table 4). If we compare muscle quality parameters (echogenicity and X-Y index), we find a negative correlation between these parameters and resistance, reactance, and phase angle (Table 4).

### 3.5. Relationship of Nutritional Ultrasonography with Morbidity

We compared the differences in ultrasonography parameters between those who suffered complications and those who did not. There were no differences between admitted patients and those who were not. There were no differences between patients who went to emergency services. Nevertheless, patients who suffered death had a higher X-Y index (4.67 (1.43) vs. 3.48 (1.32); *p* = 0.02) and a higher MCRFI than those who did not (3.86 (0.70) vs. 3.29 (0.59); *p* < 0.01). In the multivariate analysis adjusted by age, the highest quartile of the X-Y index has more risk of death OR: 4.54 CI95% (1.11–18.47); *p* = 0.03 (Figure 5).

## 4. Discussion

Nutritional ultrasonography is a novel technique that allows us to measure muscle mass and muscle quality. This study has shown that muscle mass parameters of ultrasonography as MARFI are higher in patients with sarcopenia and have a positive correlation with parameters of body composition like ASMI, MMI and phase angle. On the other hand, muscle quality parameters like muscle echogenicity and X-Y index show differences in strength criteria from sarcopenia (dynapenia) and have a negative correlation with parameters related to muscle function like handgrip strength and phase angle and reactance from BIA.

### 4.1. Use of Nutritional Ultrasonography in Disease-Related Malnutrition

Patients analyzed had a varied distribution of pathologies that cause disease-related malnutrition, with a predominance of oncologic patients. These diseases and malnutrition can produce sarcopenia, as we have seen in 30% of patients in our sample. These results are higher than those observed in a study developed in 2021 in admitted patients, with 10.5% of patients with sarcopenia and disease-related malnutrition [23]. This difference can be related to the type of patients. In our study, patients are predominantly oncologic, while in the study referred are cardiorespiratory patients. Another study by Riesgo et al. in older patients with COVID-19 showed a higher prevalence of risk of sarcopenia due to the type of disease and method of diagnosis [24].

The body composition has differences between men and women. This condition explains the changes in anthropometry, BIA, and handgrip strength that we have seen in the function of sex [25]. Ultrasonography showed differences between genders in absolute values but did not show differences if the values were standardized by height. These data are similar to those of the study of Arts et al. that showed a difference between males and females in muscular ultrasonography [26].

### 4.2. Nutritional Ultrasonography and Diagnosis of Sarcopenia

There were age differences when we compared patients with and without sarcopenia. Primary sarcopenia is a frequent disease in patients with more than 70 years. This condition relates to the reduction from 3 to 8% of muscle mass each decade since the age of 30, more marked in patients with more than 60 years [27]. DRM is related to sarcopenia, but the association with age can increase the risk of this pathology. A Pekin Union Medical College hospital study showed that the patients with risk of malnutrition and sarcopenia had a higher age than those who do not have sarcopenia, as we have reported in our study [28]. In our study, the correlation analysis pointed out that the measures from ultrasonography have values related to loss of mass and quality in relation to the increase in age. These alterations in ultrasonographic parameters are related to a decline in function. This condition has been seen in community patients with older age in a study from Albacete, where dynapenia had a higher prevalence in patients over 75 years (59.7% vs. 35.7%) [29].

Sarcopenia is defined as reduced mass and function of muscle. The usefulness of ultrasonography in the diagnosis and staging of sarcopenia can be used mostly to evaluate muscle mass. We have seen low values in structural measures of the muscle as MARF, MARFI and Y axis. The use of ultrasonography has been planted in patients with primary sarcopenia in older adults [30], but the use in secondary sarcopenia and disease-related malnutrition is still unclear. Some studies have shown low values of muscle mass measured by ultrasonography, such as Sánchez-Torralvo et al., in patients with cistyc fibrosis [31]. Another study in patients from the internal medicine department of the University Hospital of Siena showed significantly lower values of muscle thickness measured by ultrasonography in patients with sarcopenia [32]. However, if we compared the values of nutritional ultrasonography related to the low muscle mass criterion, we did not find any difference. This situation can be produced in relation to conditions that can increase the mass of muscle but decrease the function of myoesteatosis or inflammation. In a study by Bot et al. in patients with end-stage liver disease, low SMI was not related to muscle function in a 6-min walking distance, but myoesteatosis showed a relation to an altered 6-min walking distance [33].

The parameters of muscle quality as an X-Y index showed differences in sarcopenia. This marker indicates the relationship between transversal and anteroposterior axis. Low values relate to better muscle quality due to the predominance of the Y axis, which demonstrates a harder muscle. Considering only the dynapenia criterion, we observe differences in the X-Y index and echogenicity. These characteristics of muscle can help us to evaluate muscle quality and function. Muscle echogenicity has shown an inverse relationship with muscle strength, and it is related to density by CT [34].

### 4.3. Nutritional Ultrasonography in Morphofunctional Assessment

Morphofunctional assessment of DRE uses techniques of intake evaluation, body composition, muscle function and biochemical parameters to carry out a global approach to nutritional assessment. Ultrasonography plays an important role in this nutritional assessment. It is necessary to know the relationship between nutritional ultrasonography and other components of morphofunctional assessment.

Femoral muscle ultrasonography can be adequate to assess muscle mass compared to CT at the third lumbar vertebra (CT L3 MM), as described in a study by Arai et al. developed in Intensive Care Unit Admission patients. This study reported a r = 0.48 for rectus femoris, which had the discriminative power to assess low muscularity [35]. Another study by Fischer et al. observed that ultrasound measures at the tight can predict CT L3 MM in different populations with non-critical illness [16]. In our study, the assessment of muscle mass by ultrasonography using MARFI correlated with muscle mass measures determined by BIA like ASMI and MMI, and it correlated also with cellularity measures like phase angle determined by BIA. The evaluation of phase angle and its correlation with muscle mass by ultrasonography has been proposed in obese females [36] and oncological patients [37]. In this study (AnyVida trial), phase angle and ultrasonography were prognostic factors for 12-month mortality [37].

Muscle quality measurement of ultrasonography was assessed by echogenicity and X-Y index. These parameters correlated with muscle strength determined by handgrip strength. The quality measures from ultrasonography as echogenicity have demonstrated a relation to muscle strength, as in the study from Bunout et al., where the lowest muscle echogenicity is related to a higher quadriceps torque and a higher handgrip strength in older adults [34]. In another study by Mañago et al., echogenicity was inversely correlated with muscle strength (r = −0.46, *p* < 0.01) and power (r = −0.50, *p* = 0.006) in patients with multiple sclerosis.

Muscle ultrasonography quality parameters are also correlated with electric parameters from BIA, like reactance and phase angle that are related to body cell mass. Body composition assessed by BIA is based on electrical characteristics of the human body to estimate components such as muscle mass, hydration, or fat mass. However, in BIA, the direct measure from electric parameters can help us know body cell mass as a body composition variable and functional parameters. The electrical parameters from BIA can be related to disease-related malnutrition and body function or inflammation and are related to disease prognosis [20]. Correlation between ultrasound quality measures and electrical parameters from BIA leads to nutritional ultrasonography as a useful determination of muscle function, body function and disease prognosis. Phase angle has demonstrated the relationship with muscle mass and density studied with TC in a study from Gen et al. [38]; in this study, in elderly patients, lower values of phase angle are associated with low density of muscle determined by CT [38]. Another study by Bourgeois et al. showed a relationship between muscular echogenicity and phase angle in healthy individuals [39].

### 4.4. Nutritional Ultrasonography and Complications in DRE

Muscle mass parameters (MARF, muscle thickness) in ultrasonography have a relationship with prognosis in some pathologies in acute and chronic patients. A study from Málaga showed that muscle thickness is a prognostic factor for mortality in patients with cancer [37]. A systematic review conducted by Casey et al. demonstrated that cross-sectional area and muscle thickness are associated with readmission, length of stay and survival; it was done with 37 studies (22 of them are in patients in ICU) [40]. Our study did not show differences in muscle mass parameters except MCRF. Nevertheless, this parameter could have more of a relationship with muscle quality than muscle mass. Muscle size can be influenced by myoesteatosis and edema with higher values. On the other hand, the high variability of pathologies analyzed can interfere with no differences in the events analyzed.

Muscle quality parameters showed differences in the X-Y index and MCRF for mortality. X-Y index can offer us an information about muscle stiffness that cannot be done by other measures. Casey’s systematic review showed the relationship between muscle quality parameters such as echogenicity and prognosis. Conversely, muscle thickness is related to Y-axis size, which is also associated with a patient’s prognosis [40]. Echogenicity did not show differences in prognosis, but it could be related to variability in the type of patients and its relationship with hydration and the effect of treatment of the primary disease [41].

### 4.5. Strengths and Limitations

The main strength of this study is the use of a novel technique, such as nutritional ultrasonography, in a large sample of patients diagnosed with disease-related malnutrition. This condition can help us to understand the behavior of this diagnostic method in ill patients. On the other hand, this is a study in community patients. Most of the studies done with ultrasonography are in critical or non-critic hospitalized patients. At last, using the technique inside a morphofunctional assessment planning allows us to better understand the utility of ultrasonography in studying muscle mass and function of patients with DRM.

The limitations of this study were the selection of different pathologies which cause DRM are associated with a high variability in morphofunctional assessment to find differences but also gives more statistical power to the differences obtained. The lack of cutoff points to standardize ultrasonography prevents us from evaluating sarcopenia or evaluation of prognosis. Using a correlation test limits our comparison to one with the techniques, and the lack of gold standard techniques such as CT or MRI hinders an adequate validation of the test. The age of patients can interfere with an adequate interpretation of data related to the influence of age and disease over muscle mass and function.

### 4.6. Future Lines of Investigation

Nutritional ultrasonography is an emerging technique for nutritional assessment of patients since it is an inexpensive and easy-to-perform method. The morphofunctional assessment associated with ultrasonography could help us to make an easy diagnosis and follow-up of sarcopenia, disease-related malnutrition and its treatment. However, scientific evidence on disease-related malnutrition is still scarce. It is needed the categorization of cutoff points to help in the diagnosis of nutrition-related pathologies (sarcopenia and DRM). On the other hand, validation of nutritional ultrasonography is needed in the pathologies that cause sarcopenia and disease-related malnutrition. It is important to consider ultrasonography as a method to evaluate muscle mass and quality and standardize the technique to determine the measurements of variables and the most adequate muscle to use in each pathology.

## 5. Conclusions

In community patients with DRM, the prevalence of sarcopenia was 33.3%. This prevalence was superior in women than men. In patients with sarcopenia, muscle mass parameters determined by nutritional ultrasonography of the rectus femoris muscle (muscle area, muscle thickness (Y axis)) are lower than in patients without sarcopenia. Muscle quality parameters (X-Y index) showed the worst values in patients with sarcopenia; echogenicity only showed differences (higher values) in patients with dynapenia criterion of sarcopenia.

Muscle mass ultrasonography parameters were correlated with electrical parameters (resistance and phase angle), and estimated muscle parameters (ASMI, MMI) were assessed by BIA. Muscle quality parameters (echogenicity and X-Y index) had a higher correlation with electric parameters from BIA than muscle mass parameters; they correlated with muscle strength assessed by handgrip strength. Ultrasonography X-Y index (highest quartile) is associated with an increase in the risk of mortality in patients with disease-related malnutrition assessed by ultrasonography.

Muscle mass assessment by ultrasonography is a good and easy method to evaluate muscle mass and quality in patients with disease-related malnutrition. We need to develop studies to complete the evidence about this technique, standardize it, and integrate it into usual clinical practice to diagnose disease-related malnutrition and monitor medical nutrition therapy.

## Figures and Tables

**Figure 1 nutrients-15-03923-f001:**
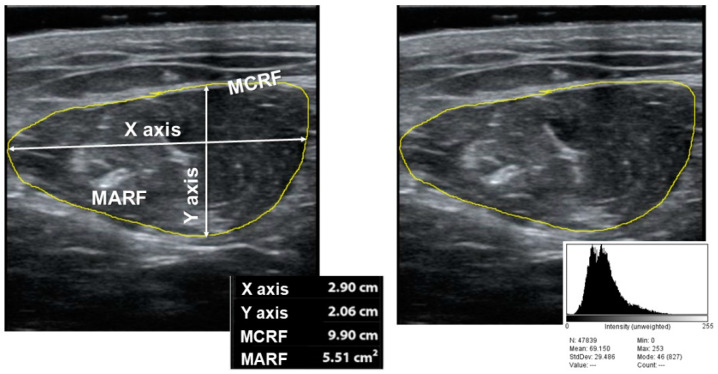
Parameters of muscle ultrasonography of quadriceps rectus femoris ((**right**) muscle mass measures; (**left**) echogenicity). MARF: Muscular Area Rectus Femoris; MCRF: Muscular Circumference Rectus Femoris.

**Figure 2 nutrients-15-03923-f002:**
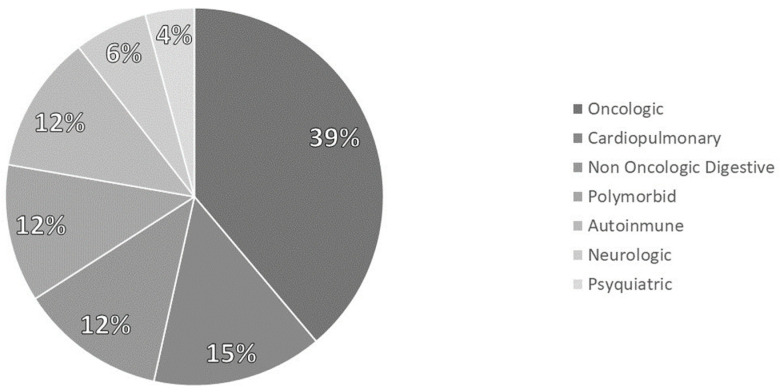
Distribution of pathologies.

**Figure 3 nutrients-15-03923-f003:**
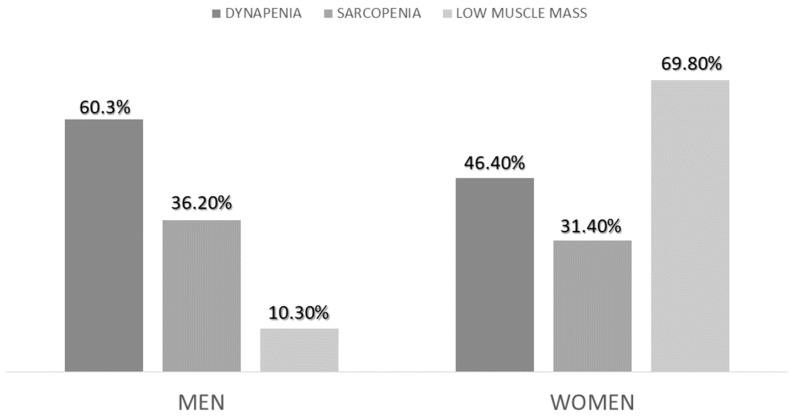
Differences in the diagnosis of sarcopenia, low muscle mass and dynapenia between sexes.

**Figure 4 nutrients-15-03923-f004:**
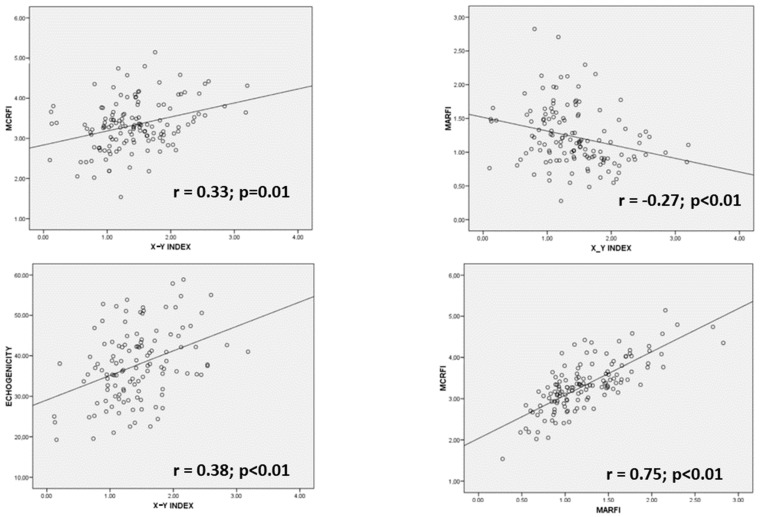
Regression graphics comparing ultrasonography variables. MARFI: muscle area of rectus femoris index; MCRFI: muscle circumference of rectus femoris index; X: transversal axis; Y: anteroposterior axis.

**Figure 5 nutrients-15-03923-f005:**
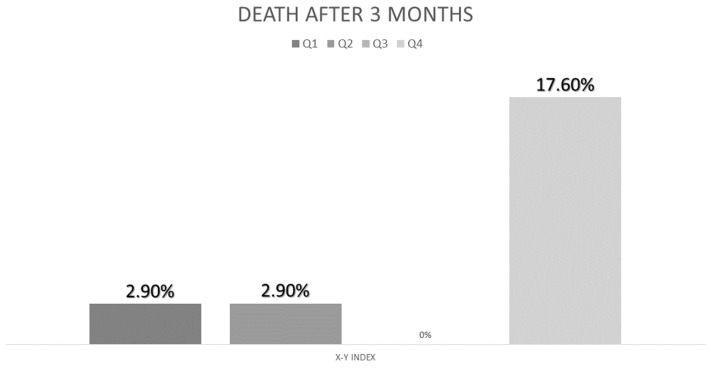
Percent of deaths related to quartile of X-Y index.

**Table 1 nutrients-15-03923-t001:** Morphofunctional assessment variables and differences between sexes.

	Total	Men	Women	*p*-Value
Anthropometry				
BMI (kg/m^2^)	21.79 (4.61)	23.99 (4.62)	20.31 (4.01)	<0.01
Age (years)	61.4 (17.34)	64.91 (14.70)	60.71 (18.80)	0.15
%weight loss	11.84 (9.44)	10.42 (7.38)	12.88 (10.63)	0.15
Arm Circumference (cm)	23.07 (2.98)	24.73 (2.92)	21.97 (2.47)	<0.01
Calf circumference (cm)	31.03 (3.42)	32.47 (3.69)	30.06 (2.86)	<0.01
Bioelectrical Impedanciometry				
Resistance (ohm)	595.81 (110.42)	531.03 (98.38)	638.73 (96.51)	<0.01
Reactance (ohm)	50.58 (11.76)	46.46 (12.17)	53.32 (10.69)	<0.01
Phase Angle (°)	4.86 (0.83)	4.99 (0.88)	4.78 (0.79)	0.15
ASMI (kg/m^2^)	5.88 (1.09)	6.77 (0.96)	5.30 (0.71)	<0.01
MMI (kg/m^2^)	9.69 (1.78)	10.75 (1.73)	9.02 (1.47)	<0.01
Nutritional Ultrasonography				
MCRFI (cm/m^2^)	3.33 (0.61)	3.19 (0.63)	3.41 (0.59)	0.03
MARFI (cm^2^/m^2^)	1.21 (0.43)	1.22 (0.51)	1.21 (0.38)	0.81
X-Y index	3.56 (1.35)	3.46 (1.37)	3.61 (1.35)	0.54
Echogenicity (%)	36.68 (9.70)	32.79 (9.42)	39.16 (9.09)	<0.01
Muscle Strength				
Handgrip strength (kg)	20.28 (7.57)	24.82 (7.93)	17.15 (0.59)	<0.01

BMI: body mass index; ASMI: appendicular skeletal mass index; MMI: muscle mass index; MCRFI: muscle circumference rectus femoris index (cm/m^2^); MARFI: muscle area rectus femoris index (cm/m^2^). X: transversal rectus femoris axis; Y: anteroposterior rectus femoris axis.

**Table 2 nutrients-15-03923-t002:** Morphofunctional assessment variables related to diagnosis of sarcopenia.

	Sarcopenia	No Sarcopenia	*p*-Value
SEX (%M/%W)	36.2%/32.1%	63.8%/67.9%	0.72
Anthropometry			
BMI (kg/m^2^)	20.07 (3.49)	22.73 (4.89)	<0.01
Age (years)	67.92 (13.56)	59.62 (18.59)	<0.01
%weight loss	13.83 (11.14)	10.76 (8.35)	0.09
Arm Circumference (cm)	22.31 (2.08)	23.49 (3.28)	0.03
Calf circumference (cm)	29.57 (2.69)	31.83 (3.53)	<0.01
Bioelectrical Impedanciometry			
Resistance (ohm)	641 (0.49)	569.53 (110.39)	<0.01
Reactance (ohm)	50.06 (9.07)	50.94 (13.02)	0.68
Phase Angle (°)	4.47 (0.79)	5.09 (0.84)	<0.01
ASMI (kg/m^2^)	5.40 (0.76)	6.16 (1.13)	<0.01
MMI (kg/m^2^)	8.69 (1.29)	10.26 (1.75)	<0.01
Nutritional Ultrasonography			
MCRFI (cm/m^2^)	3.31 (0.55)	3.34 (0.64)	0.82
MARFI (cm^2^/m^2^)	1.09 (0.39)	1.27 (0.45)	0.02
X-Y index	4.12 (1.28)	3.29 (1.32)	<0.01
Echogenicity (%)	38.13 (10.72)	36.07 (9.12)	0.27
Muscle Strength			
Handgrip strength (kg)	15.07 (5.85)	22.94 (6.96)	<0.01

BMI: body mass index; ASMI: appendicular skeletal mass index; MMI: muscle mass index; MCRFI: muscle circumference rectus femoris index (cm/m^2^); MARFI: muscle area rectus femoris index (cm/m^2^). X: transversal rectus femoris axis; Y: anteroposterior rectus femoris axis.

**Table 3 nutrients-15-03923-t003:** Differences in morphofunctional assessment variables in the function of components of sarcopenia (dynapenia and low muscle mass).

	Dynapenia	No Dynapenia	*p*-Value	Low Muscle Mass	No Low Muscle Mass	*p*-Value
Anthropometry						
BMI (kg/m^2^)	22.29 (4.69)	21.34 (4.55)	0.22	19.76 (3.68)	25.61 (3.69)	<0.01
Age (years)	68.19 (14.27)	56.15 (18.52)	<0.01	59.66 (18.33)	67.56 (14.08)	<0.01
%weight loss	11.93 (10.01)	11.52 (8.78)	0.81	13.23 (9.68)	9.55 (8.64)	0.03
Arm Circumference (cm)	23.53 (3.01)	22.64 (2.89)	0.08	21.94 (2.39)	25.16 (2.83)	<0.01
Calf circumference (cm)	30.91 (3.58)	31.23 (3.28)	0.57	29.87 (2.99)	33.22 (3.12)	<0.01
Bioelectrical Impedanciometry						
Resistance (ohm)	589.58 (110.45)	597.81 (108.62)	0.64	647.66 (89.31)	499.36 (75.92)	<0.01
Reactance (ohm)	47.37 (10.16)	54.17 (12.53)	<0.01	53.51 (11.21)	45.14 (10.88)	<0.01
Phase Angle (°)	4.61 (0.74)	5.17 (0.81)	<0.01	4,71 (0.77)	5.14 (0.88)	<0.01
ASMI (kg/m^2^)	5.94 (1.13)	5.87 (1.03)	0.68	5.35 (0.71)	6.88 (0.98)	<0.01
MMI (kg/m^2^)	9.63 (1.84)	9.85 (1.69)	0.47	8.77 (1.21)	11.40 (1.36)	<0.01
Nutritional Ultrasonography						
MCRFI (cm/m^2^)	3.31 (0.61)	3.36 (0.62)	0.62	3.27 (0.61)	3.43 (0.61)	0.13
MARFI (cm^2^/m^2^)	1.15 (0.45)	1.29 (0.41)	0.04	1.17 (0.42)	1.30 (0.45)	0.08
X-Y index	3.76 (1.41)	3.35 (1.28)	0.08	3.69 (1.30)	3.31 (1.43)	0.12
Echogenicity (%)	38.70 (10.35)	34.59 (8.49)	0.02	34.94 (9.58)	39.83 (9.21)	<0.01
Muscle Strength						
Handgrip strength (kg)	16.02 (6.02)	24.91 (6.29)	<0.01	19.78 (7.72)	21.19 (7.28)	0.29

BMI: body mass index; ASMI: appendicular skeletal mass index; MMI: muscle mass index; MCRFI: muscle circumference rectus femoris index (cm/m^2^); MARFI: muscle area rectus femoris index (cm/m^2^). X: transversal rectus femoris axis; Y: anteroposterior rectus femoris axis.

**Table 4 nutrients-15-03923-t004:** Correlation of ultrasonography with parameters of morphofunctional assessment.

	Echogenicity	Marfi	X-Y Index
Arm Circumference (cm)	r = 0.05; *p* = 0.55	r = 0.05; *p* = 0.55	r = −0.03; *p* = 0.75
Calf Circumference (cm)	r = 0.07; *p* = 0.41	r = 0.13; *p* = 0.12	r = −0.04; *p* = 0.62
ASMI (kg/m^2^)	r = −0.05; *p* = 0.56	r = 0.17; *p* = 0.04 *	r = −0.11; *p* = 0.19
MMI (kg/m^2^)	r = −0.03; *p* = 0.76	r = 0.25; *p* < 0.01 *	r = −0.23; *p* < 0.01 *
Resistance (ohm)	r = −0.03; *p* = 0.71	r = −0.17; *p* = 0.04 *	r = −0.03; *p* = 0.74
Reactance (ohm)	r = −0.21; *p* = 0.02 *	r = 0.12; *p* = 0.15	r = −0.31; *p* < 0.01 *
Phase Angle (°)	r = −0.23; *p* = 0.01 *	r = 0.35; *p* < 0.01 *	r = −0.42; *p* < 0.01 *
Handgrip Strength (kg)	r = −0.36; *p* < 0.01 *	r = 0.13; *p* = 0.13	r = −0.18; *p* = 0.04 *

MARFI: Muscular Area of Rectus Femoris Index; ASMI: appendicular skeletal muscle index; MMI: muscle mass index, * *p* < 0.05.

## Data Availability

Data is unavailable due to privacy or ethical restrictions.

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
