# Peer review of "Nutritional Ultrasonography, a Method to Evaluate Muscle Mass and Quality in Morphofunctional Assessment of Disease Related Malnutrition"

_nutrients, 2023, doi:10.3390/nu15183923_

Round 1

Reviewer 1 Report

The work is interesting and in line with current times. The text contains many errors. If all the authors reread the manuscript before submitting it, why didn’t you notice the errors?

I ask the authors to check and correct the following:

Line 21/23: You have to revise the text in round brackets; it is not clear since there are currently two parenthesis that are not closed.

Line 24: 61.4(17.34) years. = 61.4 (17.34) years. or 61.4 +- 17.34 years. / Patients with sarcopenia(S) = Patients with sarcopenia (S)

Line 25/25: You have to rewrite everything correctly. "had lower values of MARFI [S: 1.09 (0.39) cm2/m2; NoS: 1.27 (0.45), p=0.02); Y axis (S: 0.88 (0.27); 25 NoS: 1.19(0.60), p<0.01); and X—Y index (S: 1.52 (0.61); NoS: 1.30 (0.53), p<0.01]"

Abstract: You must always use the same format to indicate and report data. For example, you sometimes write X-Y index, other times X - Y index, other times X_Y index; it always shows the same way.

Line 36: Keywords: You must correct it: Nutritional Ultrasonography; disease-related malnutrition; morphofunctional assessment; echogenicity

Line 70/73: You have to correct the text, DEXA is considered the gold standard for fat mass, not for muscle mass. As you put it, CT, DEXA and MRI are all three gold standards.

Line 81: I think we can also say that we must have trained personnel capable of performing the ultrasound method and managing the data on the software.

Line 83: Fischer et al. (year).

Line 84/92: On the other hand, = You write it twice and find a synonym.

Line 120: You eliminate the double space. 

Line 120/123: You need to rewrite it better.  Acronyms should not be invented if they are already well described in the literature and accepted by the scientific world. Your %PWL is "Percentage Total Weight Loss [%TWL]" in the literature. You check it and correct it in all tests of paper. In the formula you wrote we are missing a piece; if it is a percentage, there goes the x100. Better check the formula and the data count. The circumferences of the middle arm and calf, according to which protocol were detected? Did you use the ISAK protocol? The data comes from how many measurements? Who performed the measurements was always the same operator? What instrument did you use? Have you considered the formulas for the correct/adjusted circumferences?

Line 135: You correct PhA=(X/R) * 180º/ ).

Line 156: You check and correct ", Figure 1." Figure 1 is called Figure 2 in the test. 

You have to fix all the "apices" of the text.

Line 165: [18]. = After, you click enter.

Line 178: You write Student's t-test, Mann–Whitney U test 

Line 190: Figure 2

Line 194: 45.8%

Line 195: 51.4%. You must check all the numbers you have written. They must all have the English format, with the point, not the comma.

Line 197: Figure 3 = You must check all the numbers you have written. They must all have the English format, with the point, not the comma.

Line 201: Table 1

Line 207: You must check all the numbers you have written. They must all have the English format, with the point, not the comma.

Line 214/216: Table 2

Table 2: You must check all the numbers you have written. They must all have the English format, with the point, not the comma.

Line 225/228: Table 3

Line 240: Figure 4

Line 253/255: Table 4

Table 4: This is the first time you’ve used brachial girth. The term girth is better than circumference.

Line 266: Figure 5

Figure 5: You must check all the numbers you have written. They must all have the English format, with the point, not the comma.

You have to look at the bibliographical references again, you will notice that there are formatting errors. Check and correct the entire bibliography list.

Author Response

REVIEWER 1:

Thank you very much for your comments, we think it should enrich the content of the manuscript. We answer to all your questions.

Line 21/23: You have to revise the text in round brackets; it is not clear since there are currently two parenthesis that are not closed.

Response: We have added the parenthesis.

Line 24: 61.4(17.34) years. = 61.4 (17.34) years. or 61.4 +- 17.34 years. / Patients with sarcopenia(S) = Patients with sarcopenia (S)

Response: We have corrected the typography.

Line 25/25: You have to rewrite everything correctly. "had lower values of MARFI [S: 1.09 (0.39) cm2/m2; NoS: 1.27 (0.45), p=0.02); Y axis (S: 0.88 (0.27); 25 NoS: 1.19(0.60), p<0.01); and X—Y index (S: 1.52 (0.61); NoS: 1.30 (0.53), p<0.01]"

Response: We have added bracketts.

Abstract: You must always use the same format to indicate and report data. For example, you sometimes write X-Y index, other times X - Y index, other times X_Y index; it always shows the same way.

Response: We have unified all the therms.

Line 36: Keywords: You must correct it: Nutritional Ultrasonography; disease-related malnutrition; morphofunctional assessment; echogenicity

Response: We have deleted the numbers after keywords.

Line 70/73: You have to correct the text, DEXA is considered the gold standard for fat mass, not for muscle mass. As you put it, CT, DEXA and MRI are all three gold standards.

Response: You are right we are talking about muscle, and we have included a incorrect technique as DEXA. We have changed it. Line 70-73: “There are techniques not accurate as anthropometry with perimeters or with estimative equations based on bioimpedanciometry. Besides, there are some tests like computer-ized tomography (CT) or magnetic resonance imaging (MRI) considered gold standard but more expensive and not feasible in routine clinical practice [13].”

Line 81: I think we can also say that we must have trained personnel capable of performing the ultrasound method and managing the data on the software.

Response: Thank you very much for the appreciation. We have added your suggestion to the manuscript. Line 81-83: “ Finally, this technique needs trained personnel capable of performing this ultrasound method and manage the data on the software[15].”

Line 83: Fischer et al. (year).

Response: We have added the year to the cite.

Line 84/92: On the other hand, = You write it twice and find a synonym.

Response: We have changed it as you ask. Line 94-95: “Besides, the follow-up of changes in ultrasonography can help to monitor the effect on muscle of nutritional treatment.”

Line 120: You eliminate the double space. 

Response: We have corrected it.

Line 120/123: You need to rewrite it better.  Acronyms should not be invented if they are already well described in the literature and accepted by the scientific world. Your %PWL is "Percentage Total Weight Loss [%TWL]" in the literature. You check it and correct it in all tests of paper. In the formula you wrote we are missing a piece; if it is a percentage, there goes the x100. Better check the formula and the data count. The circumferences of the middle arm and calf, according to which protocol were detected? Did you use the ISAK protocol? The data comes from how many measurements? Who performed the measurements was always the same operator? What instrument did you use? Have you considered the formulas for the correct/adjusted circumferences?

Response: We have changed the acronyms of TWL as you ask. We have completed the protocol of anthropometry measurements. Line 122-129: Anthropometry:  the anthropometric variables measured were weight (kg); height (m); body mass index (BMI) as weight/height*height (kg/m2); percentage of weight loss (%TPWL): (Usual weight (kg) – Actual weight (kg))/Usual weight (kg)*100).; arm cir-cumference (AC)Braquial girth (BG) (cm) and calf circumference (CC) cm were meas-ured using International Society for the Advancement Kinanthropometry (ISAK) pro-tocol. The braquial girth was made at middle point between acromium and radium head with relaxed arm. The calf circunference was made with the patient sitted in the max-imum perimeter point between knee and ankle [19].  

Line 135: You correct PhA=(X/R) * 180º/ ).

Response: We have corrected it, we have added pi symbol.

Line 156: You check and correct ", Figure 1." Figure 1 is called Figure 2 in the test. 

Response: We have corrected it and all throughout text.

You have to fix all the "apices" of the text.

Line 165: [18]. = After, you click enter.

Response: We have added enter.

Line 178: You write Student's t-test, Mann–Whitney U test 

Response: We have changed these therms.

Line 190: Figure 2

Response: We have corrected it and all throughout text.

Line 194: 45.8%

Response: We have corrected it and changed all the commas by points in the manuscript.

Line 195: 51.4%. You must check all the numbers you have written. They must all have the English format, with the point, not the comma.

Response: We have corrected it and changed all the commas by points in the manuscript.

Line 197: Figure 3 = You must check all the numbers you have written. They must all have the English format, with the point, not the comma.

Response: We have corrected it and changed all the commas by points in the manuscript.

Line 201: Table 1

Response: We have corrected it and all throughout text.

Line 207: You must check all the numbers you have written. They must all have the English format, with the point, not the comma.

Response: We have corrected it and changed all the commas by points in the manuscript.

Line 214/216: Table 2

Response: We have corrected it and all throughout text.

Table 2: You must check all the numbers you have written. They must all have the English format, with the point, not the comma.
Response: We have corrected it and changed all the commas by points in the manuscript.

Line 225/228: Table 3

Response: We have corrected it and all throughout text.

Line 240: Figure 4

Response: We have corrected it and all throughout text.

Line 253/255: Table 4

Response: We have corrected it and all throughout text.

Table 4: This is the first time you’ve used brachial girth. The term girth is better than circumference.

Response: We have changed the term arm circumference by brachial girth throughout the text.

Line 266: Figure 5

Response: We have corrected it and all throughout text.

Figure 5: You must check all the numbers you have written. They must all have the English format, with the point, not the comma.

Response: We have corrected it and changed all the commas by points in the manuscript.

You have to look at the bibliographical references again, you will notice that there are formatting errors. Check and correct the entire bibliography list.

Response: We have checked and corrected all the references.

Reviewer 2 Report

The present article presents the usefulness of a new approach in malnutrition induced sarcopenia evaluation, namely muscular ecography. As life span tends to grow, sarcopenia becomes a reality in many ageing people, with or without associated diseases. In consequence, any technique that facilitates its evaluation, especially if it is cost efective, is welcomed. Here we see a research concerning disease related sarcopeny, taking under evaluation a certain tipe of sonography, the nutritional one, measuring mass and quality of muscles. Researchers compare it with other well established tools like bioimpedance, somatometry and muscle dynamometry. The article describes in detail the theoretical and material apparatus used, the statistical tests and the results. Most of the results confirm correlations between findings in all tests applied. Visual material is helpful, references are correctly cited and limitations are clearly presented. The article is very dense, so I suggested a graphical abstract. 

Conclusions are also very interesting, underlining that the present research has no definitive findings but it opens a path for further research. 

This is an overall interesting article and I congratulate the authors for using such novel approach to malnutrition evaluation. I suggest to add a graphic abstract, or some kind of simplified scheme of the research, in order to facilitate the understanding of the results. Also, since just correlation tests were used, can you please present limitations of this type of tests. 

Author Response

Thank you very much for the comments. We have tried to change the parts that you suggest:

  • We have added a Graphical Abstract.
  • We have added the limitations that you suggest.
    • Line 411-413: “The use of a correlation test limits us a comparison one to one with the techniques, and the lack of gold standard techniques as CT or MRI hinders an adequate validation of test.”

Round 2

Reviewer 1 Report

Your auditor is an ISAK 3 instructor. You’re claiming you used the ISAK method. If you have used the ISAK method you know that the terminology and anatomical points are well-coded and standardized. The term brachial girth does not exist in ISAK terminology. None of the authors on the ISAK website is an ISAK-certified anthropometrist (https://www.isak.global/MemberList/Find). The circumference of the calf with the ISAK method is measured with the subject in an anthropometric position (standing, not sitting). When you write, it suggests: 1) that you did not really use the ISAK method; 2) that you used the ISAK method incorrectly. In scientific articles, it is very serious to make false statements. So, I ask you if you really used the ISAK method (in this case incorrectly) or if you used another anthropometric method. For example, according to this method, the subject is sitting, but according to the ISAK method, the subject is standing.

chrome-extension://efaidnbmnnnibpcajpcglclefindmkaj/https://www.cdc.gov/nchs/data/nhanes/bm.pdf

Who performed the anthropometric measurements was trained in anthropometry? Did he have any skills? How many measurements did he take? Have the measurements always been taken by the same operator? On which side of the body were the measurements made (right or left)?

Author Response

Thank you for your comments. First place, I would like to apologize for the not use of adequate correction of therms. I misunderstood the correction in this point, and we have made some mistakes related to the use of english. I would like to explain and correct all possible mistakes in text.

Your auditor is an ISAK 3 instructor. You’re claiming you used the ISAK method. If you have used the ISAK method you know that the terminology and anatomical points are well-coded and standardized. The term brachial girth does not exist in ISAK terminology. None of the authors on the ISAK website is an ISAK-certified anthropometrist. (https://www.isak.global/MemberList/Find).

In relation to the use of brachial girth, we have changed as we understood that you ask us to change it. “Table 4: This is the first time you’ve used brachial girth. The term girth is better than circumference.” Sorry for the mistake, probably we misunderstood this request. We have changed it to arm circumference through text as it is states in NHANES protocol you mentioned.

None of us is a certified anthropometrist from ISAK. We have used a modified ISAK method from “Higher Sports Council” of Spain (reference 19 of the article. ISBN: 978-84-7949-220-5) by a dietitian formed in anthropometry measurements but not certified by ISAK.

The circumference of the calf with the ISAK method is measured with the subject in an anthropometric position (standing, not sitting). When you write, it suggests: 1) that you did not really use the ISAK method; 2) that you used the ISAK method incorrectly. In scientific articles, it is very serious to make false statements. So, I ask you if you really used the ISAK method (in this case incorrectly) or if you used another anthropometric method. For example, according to this method, the subject is sitting, but according to the ISAK method, the subject is standing.

chrome-extension://efaidnbmnnnibpcajpcglclefindmkaj/https://www.cdc.gov/nchs/data/nhanes/bm.pdf

Sorry again for the misunderstanding, we have used ISAK method planted in the document: “Variables antropométricas de la población deportista española. Madrid: Consejo Superior de Deportes, Servicio de Documentación y Publicaciones; 2012” but it seems that we have some mistakes doing it, so we have made it in an incorrect manner.

We have also made a mistake when we have described the measurement of calf circumference as we did it with the patient in a stand position, not sitted. This was my neglect after translation from Spanish. Sorry about that.

Who performed the anthropometric measurements was trained in anthropometry? Did he have any skills?

The person who did anthropometry was a dietitian-nutritionist licensed by University of Navarra and formed in anthropometric measurement. She had skills on nutritional assessment and anthropometry.

How many measurements did he take? Have the measurements always been taken by the same operator? On which side of the body were the measurements made (right or left)?

It was taken one measure at right member (arm and calf). The measurements were always taken by the same operator.

We have changed the paragraph entirely to adapt the request and we have changed the therm girth to “arm circumference” through text as well in NHANES document you have sent.

Line 121-133: “Anthropometry: the anthropometric variables measured were weight (kg); height (m); body mass index (BMI) as weight/height*height (kg/m2); percentage of weight loss (%TWL): (Usual weight (kg) – Actual weight (kg))/Usual weight (kg)*100). Arm Circumference (AC) (cm) and calf circumference (CC) cm were measured using guideline of “Anthropometric variables of the Spanish sports population” that use a modified version of International Society for the Advancement Kinanthropometry (ISAK) protocol.  The arm circumference was made at middle point between acromium and radium head with relaxed arm. The calf circunference was made with the patient standing in the maximum perimeter point between knee and ankle [19]. It was taken one measure at right member (arm and calf). The person who did anthropometry was a dietitian-nutritionist formed in anthropometric measurement with skills on nutritional assessment and anthropometry. The measurements were always taken by the same operator.”

I would like to apologize again for the mistakes in the definition of the protocol. We have tried to solve it in the better way.

If you consider that we must do any other change or is necessary any other clarification about the manuscript, please let us know.

Reviewer 2 Report

You covered all the requests I indicated so I consider the article publishable.

Author Response

Thank you very much for the interest.